# Association of Diabetes with Greater Mid-Term Cognitive Decline After Carotid Surgery

**DOI:** 10.3390/biomedicines13092188

**Published:** 2025-09-07

**Authors:** Ágnes Dóra Sándor, Péter Márk Sikos, Gabriel Varinot, Fotis Kallinikos, Csongor Mánfai, Mandula Ifju, Tibor Kézi, Zsófia Czinege, András Szabó, Zsuzsanna Mihály, Péter Sótonyi, Andrea Székely

**Affiliations:** 1Department of Anesthesiology and Intensive Therapy, Semmelweis University, 1085 Budapest, Hungary; 2Master’s Program in Autonomous Systems, Faculty of Informatics, Eötvös Loránd University, 1053 Budapest, Hungary; 3Department of Medicine, Université Laval, Québec, QC G1V 0A6, Canada; 4Department of Medicine, University of Patras, 26504 Patras, Greece; 5Mihály Fazekas High School, 1071 Budapest, Hungary; 6Doctoral School, Semmelweis University, 1085 Budapest, Hungary; 7Genontech Ltd., 1134 Budapest, Hungary; 8Department of Vascular and Endovascular Surgery, Semmelweis University, 1085 Budapest, Hungary; 9Intensive Care Unit, Department of Anesthesiology, Petz Aladár University Teaching Hospital, 9024 Győr, Hungary; 10Department of Oxyology and Emergency Care, Faculty of Nursing Studies, Semmelweis University, 1085 Budapest, Hungary

**Keywords:** carotid endarterectomy, cognitive function, diabetes mellitus, cerebral tissue saturation, NIRS monitoring, MACCE, survival

## Abstract

**Background/Objectives**: Long-term cognitive outcomes after carotid surgery are influenced by diabetes and intraoperative changes. We aimed to analyze the postoperative cognitive changes in diabetic patients and nondiabetic patients after carotid endarterectomy (CEA). Additionally, major cardiovascular and cerebrovascular events (MACCEs) and the incidence of mortality at two years after surgery were assessed. **Methods**: We enrolled 37 diabetic and 67 nondiabetic patients undergoing elective carotid surgery. Intraoperatively, routine monitoring was completed with NIRS (near-infrared spectroscopy) and an Entropy monitor was used for neuromonitoring. The lowest cerebral tissue saturation levels during the cross-clamp period (rSO_2lowestclamp_) and the degree of desaturation were calculated. We used MMSE (Mini-Mental State Examination) and MoCA (Montreal Cognitive Assessment) to assess cognitive function. Cognitive change was defined as one standard deviation (SD) change from the preoperative test scores. **Results**: The MMSE and MoCA were available for 103 patients at three months and for 90 patients at 12 months after discharge. Compared with nondiabetic patients, diabetic patients exhibited greater decreases in MoCA scores (*p* = 0.028 and *p* = 0.042 at the 3rd and 12th months, respectively). Cognitive improvement was lower in the DM group than in the control group at the 12th month (18.75% vs. 42.86%, respectively; *p* = 0.029). The mean rSO_2_ in the pre-clamping period (67.4% vs. 74.6% in diabetic and in nondiabetic patients, respectively; *p* = 0.011) was lower in diabetic patients. Furthermore, MACCEs at the 24th month were observed at a higher rate in diabetic patients (*p* = 0.040). **Conclusions**: Diabetic patients demonstrated greater risks for cognitive decline, MACCEs, and mortality at two years after surgery.

## 1. Introduction

Diabetes mellitus is a chronic metabolic disorder that affects 537 million adults worldwide [1]. Among several long-term complications of this disease, there is a well-established association between diabetes and carotid atherosclerosis [2]. Carotid atherosclerosis (particularly extracranial internal carotid stenosis) is one of the leading causes of ischemic stroke, accounting for 8–15% of these events [3].

Given that diabetes is associated with an elevated risk for stroke and that the progression of neurologic injury represents an even worse outcome, the prevention of this development has demonstrated increased relevance [3,4,5]. Carotid endarterectomy is an effective method for preventing stroke; however, with respect to diabetic patients, there are conflicting results regarding the risks of complications overwhelming the benefits of surgery [3,6,7,8].

In addition to cardiovascular and cerebrovascular complications, a highly sensitive benchmark of the success of surgery is postoperative cognitive function, especially regarding changes in postoperative cognitive function. Postoperative cognitive decline is associated with increased mortality and morbidity and represents a significant financial burden on the healthcare system and families [9,10,11].

During carotid surgeries, alterations in cerebral perfusion play a substantial role in the postoperative change in cognitive function [12,13]. Preventing these major changes in cerebral perfusion may have a beneficial effect on cognition postoperatively. This underscores the relevance of neuromonitoring during this intervention.

According to the majority of clinical trials, diabetic patients have an elevated risk for cognitive impairment postoperatively [14]. The background of this phenomenon is complex. In addition to maintaining adequate glycemic control, the optimization of intraoperative hemodynamic parameters and the maintenance of adequate cerebral circulation during the cross-clamp period may be crucial in preventing complications and protecting the integrity of postoperative cognitive function in diabetic patients [15,16,17]. The monitoring of cerebral function is an important step in this management protocol [18]. NIRS (near-infrared spectroscopy) has been proven to be a reliable tool for monitoring cerebral perfusion [19]. A significant drop in cerebral tissue saturation may have a detrimental impact on postoperative cognition [20]. The use of an entropy monitor in tracking the depth of anesthesia has been demonstrated to reduce the risk of cognitive dysfunction [21]. The combination of these two techniques may increase patient outcomes with regard to cognitive safety.

For assessing cognitive function, several screening tools are available. Although the Mini-Mental State Examination (MMSE) is able to detect major changes in cognition, the Montreal Cognitive Assessment (MoCA) may be useful in the detection of mild cognitive changes [22,23].

The MoCA Memory Index Score (MoCA-MIS) has further enhanced of this detection ability and has been proven to be useful in distinguishing unimpaired cognition from mild impairment [24].

This study aimed to analyze the postoperative changes in cognitive function in diabetic and nondiabetic patients and to evaluate the connection between cognitive changes and cerebral tissue saturation values. The secondary aim of the study was to analyze the occurrence of two-year cerebral and cardiovascular events, as well as mortality.

## 2. Materials and Methods

Participants

This study was approved by the Ethical Committee of Semmelweis University (number: 17/2019, date: 15 February 2019) and registered on ClinicalTrials.gov (NCT03907943). Then, patients undergoing elective carotid endarterectomy at a cardiovascular center were enrolled consecutively in this study. After providing detailed information, all patients provided written informed consent to participate in the study.

All patients were asymptomatic with the severity of stenosis exceeding 70% based on the North American Symptomatic Carotid Endarterectomy Trial (NASCET) criteria. The type of surgery (eversion endarterectomy (EEA) or thromboendarterectomy (TEA)) and the use of shunt insertion were utilized based on the discretion of the surgeons. The indication for shunt placement was incomplete circle of Willis.

The anatomy of the circle of Willis was classified as being complete (no component was hypoplastic or absent) or incomplete.

The exclusion criteria included patients with symptomatic carotid artery stenosis, atrial fibrillation, preoperative presence of dementia (preoperative MMSE score < 24), and lack of consent. Between May 2019 and November 2021 (13 May 2019–4 November 2021), 129 patients were enrolled in the study.

Three patients were not able to achieve 24 points on the MMSE. In eleven cases, accurate recording via anesthesiological monitoring techniques could not be interpreted due to technical reasons.

One patient died before the first control (due to SARS-CoV pneumonia). Eleven patients did not return for our follow-up visit. Since their data were missing at random, there was no selection bias, and they could be excluded from further analysis.

Thus, 103 patients completed three-month postoperative cognitive assessments. One year after surgery, cognitive data were available for 90 patients. For the 76 patients who completed the preoperative and 3-month cognitive surveys, complete intraoperative monitoring was appropriately recorded. The flow diagram of the enrollment and follow-up details of the study are shown in Figure 1. Cognitive follow-up ended in November 2023.

During data processing, patients were divided into two groups based on the presence of diabetes mellitus. The diabetes diagnosis was based on the definition established by the WHO [25]. Hypoglycemic medications and blood glucose levels were managed according to the guidelines of the Academy of Medical Royal Colleges [26].

The diabetes medications that our patients used are summarized in Appendix A.

To predict survival, we used a recently published risk score [27]. This score was created based on the data of 24,713 patients with symptomatic carotid stenosis before carotid endarterectomy.

Parametric, nonparametric, and binary logistic regression statistical analyses were used to identify the determining factors. Factors that were significant at the 0.05 level in the multivariate regression analysis were included as predictors of the risk score.

The value of a factor’s beta coefficient indicates its weight on the score, which was calculated in the regression analysis. One point was added for every 0.25 interval.

The factors associated with a greater risk included female sex (1 point), diabetes (1 point), age (different age groups), malnutrition (BMI < 20) (2 points), a condition reaching a value of 4 on the American Society of Anesthesiology classification (2 points), and home status (1 point), as these factors had been shown to influence survival. When considering comorbidities, congestive heart failure (2 points), chronic obstructive pulmonary disease (2 points), intervention due to peripheral arterial disease (1 point), previous CEA or stent insertion (2 points), major lower extremity amputation indicated in the patients’ anamnesis (3 points), chronic renal insufficiency (2 points), the need for dialysis (4 points), and anemia (4 points) were identified as risk factors for impaired survival, in addition to the anamnesis of smoking (2 points). As the enrolled patients were symptomatic, the severity of the neurologic injury indicated by the Rankin score was emphasized in the score. 

The efficiency of the score was also tested in symptomatic patients.

The score was strongly correlated with the mortality rate of symptomatic patients who underwent carotid artery stenting (AUC: 0.70; Hosmer–Lemeshow overall accuracy: 91.3%). This score proved to be reliable in predicting survival after the intervention.

Intraoperative management

All of our patients underwent general anesthesia. The anesthesia station was GE Aisys CS2 (GE Healthcare, Madison, WI, USA).

We used propofol (2–5 mg/kg or until loss of consciousness) and fentanyl (2–10 mcg/kg) for the induction, and sevoflurane (114 patients, 1.4–1.9%) or isoflurane (12 patients, 0.9–1.2%) for the maintenance of anesthesia with fentanyl (50–100 μg or as necessary to avoid signs of pain) as an analgesic agent. Atracurium or rocuronium was applied for muscle relaxation. Patients were ventilated with 40–50% oxygen, and end-tidal carbon dioxide was kept between 33 and 40 mmHg. Standard, routine monitoring (electrocardiogram, intra-arterial blood pressure measurement, pulse oximetry, and capnography) was completed using near-infrared spectroscopy (NIRS) and entropy monitoring. To detect regional cerebral oxygenation (rSO_2_), we used an INVOS^TM^ 5100 C (Somanetics Co., Troy, MI, USA) cerebral oximeter, which continuously recorded the regional cerebral oxygen saturation throughout the entire procedure. The sampling frequency of the NIRS monitor was 0.166 Hz.

To calculate the degree of desaturation as a percentage during the cross-clamp period, we used the following formula: (rSO_2preclamp_ − rSO_2lowestclamp_)/rSO_2preclamp_ × 100.

As the baseline, we determined the median rSO2 values of the 2-min-long pre-clamping period, and the lowest rSO2 was defined as the median of 30 s of the lowest cerebral tissue saturation values during the cross-clamp period.

To monitor the depth of anesthesia, we used a GE entropy monitor to achieve and maintain the optimal level of anesthesia and avoid anesthetic overdose. The entropy values were maintained between 40 and 60 [28].

We paid special attention to maintaining hemodynamic stability. The mean arterial pressure (MAP) was maintained within the ±20% range of the preoperative level. Before and during the cross-clamp period, MAP was kept within the +0–20% range of the preoperative level. To prevent hypotension, we used intravenous norepinephrine at a dose of 0.03 µg/kg/min, which was modified by units of 0.02 µg/kg/min in cases of devi-ations from this range. Lidocaine was administered to the surgical area in cases of bradycardia. All of the patients received 2500 IU of heparin before the internal carotid artery (ICA) was clamped, which was reversed with the administration of protamine at the end of the procedure. To evaluate trends in the monitored vital parameters, the surgery was divided into three periods: pre-clamping, clamping, and post-clamping.

The pre-clamping period began with the completion of intubation and continued until the cross-clamping of the common carotid artery (CCA). The clamping period be-gan from the time of clamping of the CCA and internal carotid artery (ICA) and con-tinued until the complete removal of the clamp. Then, the post-clamping period began and continued until extubation. The data collected from the monitor were saved as .asc files and Excel files. The interventions performed by the surgeons and anesthesiologist were accurately indicated in the database.

Cognitive evaluation

For the assessment of cognitive function, we used MMSE and MoCA. The tests were scheduled to be administered one day before surgery and at 3 and 12 months after dis-charge. The tests were administered by the same physician (SA-HU710556625-01).

The assessment began with the administration of the MMSE in order to exclude those individuals who were suffering from dementia. The MMSE contains 11 major items and is useful for assessing the function of five cognition domains. Scores of 23 or lower are indicative of cognitive impairment. In our study, patients who were not able to achieve 24 points on the MMSE were excluded [29]. Cronbach’s alpha of the MMSE scores was 0.79.

The MOCA test was developed to detect mild cognitive impairment [22]. A score of 26 points or above indicates normal cognitive performance. 

A subscore of the MoCA, the MoCA Memory Index Score, can reveal deficits in encoding memory. This score ranges from 0 to 15 and weighs recall based on the cor-rectly answered categories of the patients. The number of recalled words is multiplied by 3 if they are provided in free delayed recall, by 2 if they are provided in category-cued recall, and by 1 if they are provided in multiple-choice-cued recall. The cutoff value was demonstrated to be 7 points; however, these data should be interpreted in conjunction with the total score of the MoCA test [30].

Outcome variables

Cognitive changes represented the primary outcome of this study, and the sec-ondary outcomes were the occurrence of major adverse cardiovascular and cerebro-vascular events and mortality. Postoperative cognitive change was defined as the difference between the results of the postoperative tests and the preoperative scores by one standard deviation or more (MMSE: 1.79; MoCA: 2.28; MIS: 2.55) [31]. For each test and for the MIS, cognitive improvement was defined as one SD or greater change in the score compared with the baseline. Cognitive decline was defined as a decrease in the follow-up score by a value of one SD or more. If the change in the scores remained within the +/−1 SD limits, no change in cognitive function was detected. 

The occurrences of MACCEs were registered during the follow-up period and were defined as myocardial infarction, new onset of arrhythmia, hemodynamic instability, cardiac failure, cardiac arrest, death due to cardiac causes, pulmonary embolism, in-tracerebral hemorrhage, transient ischemic attack (TIA) or stroke, and stroke-related death.

Statistical analysis

For statistical analysis, RStudio (R version 4.4.1, RStudio version 2024.12.0) and SPSS software (IBM SPSS Statistics Version 20) were used, and the figures were created via SPSS, RStudio (ggplot2 version 3.5.1) and GraphPAD Prism (GraphPAD Prism version 10.0.0., Boston, MA, USA).

The distribution of the data was determined using the Kolmogorov–Smirnov test or Shapiro–Wilk test. The data are presented as the means and standard deviations in cases with a normal distribution and as medians and interquartile ranges in cases with nonnormal distributions. Categorical variables are presented as numbers and percentages.

We used the Mann–Whitney U test and Student’s *t* test to compare groups in cases of nonnormally distributed data and normally distributed data, respectively. Discrete data were compared using Pearson’s chi-square test, whereas data containing a small number of cases were analyzed using Fisher’s exact test.

In the analysis of the intraoperative data, discrete signals were aligned to a common reference frame and normalized using third-degree spline interpolation to ensure comparability across operations and individuals. Time-alignment and Gaussian windowing were applied for NIRS signal preprocessing. Statistical analysis included hypothesis testing and LOESS regression, with appropriate tests selected based on variable distribution characteristics using R algorithms. A sliding window minimum was computed to identify the lowest sustained intervals, with the difference value calculated as the difference between the minimum and sustained minimum values.

Linear regression analyses were used to determine the associations between pre- and intraoperative factors and postoperative MOCA changes compared to the baseline. Uni- and multivariable Cox regression were used to determine the association between perioperative factors, post-discharge cognitive disorders, and 2-year MACCE and mortality. All significant risk factors were entered into the multivariable regression model.

A *p* value < 0.05 was considered statistically significant.

## 3. Results

### 3.1. Baseline Characteristics

The data of 104 patients were analyzed. The median follow-up time was 37 months (IQR: 30–40.75 months). In total, 37 patients (35.7% of the population) had diabetes, 25 (67.56%) of whom received oral hypoglycemic medications, and 12 (32.43%) patients received insulin alone or in combination with oral hypoglycemic medications.

Diabetic patients did not demonstrate significant differences in terms of coexisting diseases, with the exceptions of congestive heart failure and intervention due to peripheral arterial disease in the anamnesis. The preoperative creatinine level revealed that the occurrence of renal insufficiency was greater in the diabetic group. Preoperative cholesterol levels were lower in the diabetic group than in the control group (*p* = 0.009), whereas fasting blood glucose levels were higher in diabetic patients (*p* = 0.001). The baseline characteristics of the two groups are summarized in Table 1.

In diabetic patients, the pathology of the cerebral arteries was characterized by more severe stenosis of the contralateral carotid artery. During the operation, shunt insertion and eversion endarterectomy were more frequently used in the diabetic group. These data are presented in Table 2.

### 3.2. Cognitive Evaluation

Three months after surgery, cognitive function was assessed in 103 patients. The preoperative MMSE and MoCA scores did not differ between the groups. Three months after surgery, the change in the MMSE score (*p* = 0.936) was not significant; however, the change in the MoCA score was significant between diabetic and nondiabetic patients (*p* = 0.028). The data are shown in Figure 2. Compared with the score at baseline, individual decline in the MIS score was significantly higher in the diabetic group than in the nondiabetic group at three months after surgery.

One year after surgery, individual changes in the MMSE score were not significantly different between the groups (*p* = 0.921), whereas the decline in the MoCA score was greater in diabetic patients than in nondiabetic patients (*p* = 0.042). The MIS scores were significantly different in the 12th month between diabetic and nondiabetic patients (*p* = 0.042).

Cognitive improvement based on the MoCA test was less likely to occur in diabetic patients than in nondiabetic patients (*p* = 0.029) one year after surgery. Similarly, improvement based on the MIS scores was significantly greater in the nondiabetic group (*p* = 0.002) at the time of this assessment. The data are shown in Figure 3.

Univariable linear regression revealed that the change in the MoCA score at 12 months after surgery was positively associated with the median value of the lowest rSO_2_ values during the cross-clamp period and male sex. It was negatively associated with diabetes mellitus, the median value of the degree of cerebral tissue desaturation, and shunt use. The results are presented in Appendix A. Multiple linear regression revealed that diabetes mellitus, the degree of cerebral tissue desaturation, and male sex were independently associated with the 12-month MoCA change. The results of the analysis are shown in Appendix A.

### 3.3. Major Adverse Cerebral and Cardiovascular Events

No patient died within the first 30 days after surgery. During the perioperative period, MACCEs occurred in four patients, three of whom had diabetes (*p* = 0.087). Additionally, two patients experienced TIA. During the 34-month average follow-up period, eleven patients died, four of whom experienced MACCE-related events. All-cause mortality was greater in the diabetic group compared with the nondiabetic group (*p* = 0.040). During the follow-up, eleven patients experienced MACCEs, seven of whom had diabetes (*p* = 0.040). A Kaplan–Meier analysis was performed for worse postoperative outcome (survival + major cardiovascular and cerebrovascular event), and the curve is shown in Figure 4.

Using the risk score of Blecha et al., we identified 4.5 points as the threshold for survival, with a sensitivity of 72.7% and specificity of 74.2% (ROC analysis AUC: 0.848; *p* = 0.001).

In the multivariate Cox regression, postoperative cognitive decline, as measured with the MoCA (*p* = 0.018), and the risk score reported by Blecha et al. (*p* = 0.001) were independently associated with mortality and MACCEs. Stratification according to diabetes status did not affect these relationships. The results of the univariate Cox regression analysis are shown in Appendix A, and the results of the multivariate analysis are summarized in Table 3.

### 3.4. Intraoperative Parameters

Seventy-six patients had complete intraoperative registration information. Systolic arterial blood pressure (SAP) values were significantly greater in the diabetic group than in the nondiabetic group across all three surgical periods: pre-clamping (*p* = 0.036), cross-clamping (*p* < 0.001), and post-clamping (*p* = 0.034). The mean arterial pressure (MAP) was also greater in diabetic patients during the cross-clamping period (*p* = 0.001), whereas the diastolic arterial pressure (DAP) was observed to be significantly different during the cross-clamping period (*p* = 0.037). The frequencies of vasopressor use at pre-clamping and post-clamping (*p* = 0.001 and *p* = 0.023, respectively) were significantly greater among diabetic patients, whereas the total applied doses were significantly greater in all three phases of the operation (pre-clamp.: *p* = 0.005; cross-clamp.: *p* = 0.003; post-clamp.: *p* = 0.033). 

Significant differences were also observed in cerebral tissue oxygenation. Diabetic patients exhibited lower median ipsilateral rSO2 values in the pre-clamping period (67.4% vs. 74.6%, *p* = 0.011) and lower minimum values during the clamping (54.4% vs. 60.0%, *p* = 0.052) and pre-clamping (61.1% vs. 68.3%, *p* = 0.033) periods. Although the contralateral rSO2 values exhibited a similar trend, the differences were not statistically significant.

A categorical analysis revealed that during the pre-clamping period, a median rSO2 value of less than 65% was observed in 38.70% of diabetic patients and in only 15.55% of nondiabetic patients, presenting a significant difference (*p* = 0.022).

With respect to entropy, minimum state entropy (SE) was significantly lower in the post-clamping period among diabetic patients (29.1 vs. 37.6, *p* = 0.003), thereby suggesting a deeper anesthetic effect or altered brain activity. Additionally, the SE difference (mean–minimum) was significantly greater in diabetic patients than in nondiabetic patients in the post-clamping period (*p* = 0.009).

Heart rate (HR) was higher in diabetic patients during the post-clamping period than in nondiabetic patients (71.8 vs. 64.6, *p* = 0.035), which aligns with the overall observed increased hemodynamic lability. 

All intraoperative parameter comparisons are presented in Table A1 and Table A2 in the Appendix B.

The results in Table A1 and Table A2 are based on hypothesis tests and are visualized using stacked density plots in Figure 5. Each plot shows the distribution of diabetic (in red) and non-diabetic (in blue) patient groups for the respective variable. The central tendency (in this case, the median) of each group is indicated by a dashed line. The X-axes are not aligned to enhance visual clarity. The median, the minimum (calculated using a sliding window of 10 timestamps), and the difference between the minimum and the median were computed for each variable.

## 4. Discussion

We observed that the performance of diabetic patients on postoperative cognitive assessments was worse than that of nondiabetic patients; specifically, diabetic patients exhibited a greater decline in preoperative scores on the MoCA test and a lower MIS score for both the 3- and 12-month assessments after discharge than nondiabetic patients. Poor outcome at the 24th months–both MACCE and mortality together– was independently associated with cognitive decline detected at one year after surgery and the risk score developed by Blecha et al.

Diabetic patients exhibited higher blood pressure values throughout the operation; however, they required more vasopressors. They exhibited lower rSO_2_ values in the preclamping period, and the lowest rSO_2_ values were also observed to be lower in diabetic patients than in nondiabetic patients.

Postoperative decline in cognitive function demonstrates many adverse consequences, both regarding perioperative circumstances, morbidity, and mortality, and in terms of enormous emotional and financial burdens on families [9,10,11,32].

Furthermore, impairment of memory involves both difficulties in everyday activities. New onset of a decreasing tendency in MIS results may increase the possibility of conversion from mild cognitive impairment to Alzheimer’s disease [33]. In our study population, diabetic patients achieved lower scores on both of the postoperative MoCA tests. In accordance with our results, a worse cognitive performance of diabetic patients has been demonstrated in several previous trials [32,34,35].

Preservation, or ideally, improvement of cognition, is one of the most desired outcomes after carotid surgery. Intact cognition is essential to an independent and healthy life with good quality. One year after the surgery, there was a significant difference in relation to cognitive improvement with diabetic patients falling behind nondiabetic patients. Beyond intraoperative factors, chronic hyperglycemia may worsen this outcome [36,37]. As glycated hemoglobin was observed to be elevated in a substantial proportion (70%) of our diabetic patients, this scenario may have exerted a negative influence on postoperative cognitive performance. This raises the question of whether preoperative optimization of metabolic state, tight perioperative gylcemic control, and accurate risk factor assessment and treatment could eliminate this difference, as these interventions have been proven beneficial in reducing postoperative complications [6,38].

Worse cognitive performance was only observed in relation to the MoCA and its subscore (the MIS), which is in accordance with the well-known feature of the MoCA test regarding the notion that it is more sensitive for detecting mild changes in cognitive function [22,39,40]. The MMSE cannot detect these changes and appears to be less effective for use in follow-ups.

Cerebral tissue saturation absolute values and relative changes have been linked to the postoperative modification of cognition [20,41,42]. A lower absolute rSO_2_ value has been demonstrated to be an independent predictor of postoperative delirium and complications in patients receiving valvular heart and thoracic surgery [43]. In another study, the threshold for elevated risk for postoperative morbidity was determined at rSO_2_ values of 65% or lower [44]. In our population, a median value of 65% or less in cerebral desaturation was observed in approximately 15% of the nondiabetic patients, whereas it was detected in approximately 39% of the diabetic patients. In a recent study, the normal value of average cerebral tissue saturation was observed to be 67.6%, with the lowest threshold of 56% being observed in healthy volunteers [45]. In our cohort, the rSO_2_ values measured during the preclamping period were demonstrated to be 67.4% in diabetic patients and 74.6% in nondiabetic patients. The lowest rSO2 value measured during the cross-clamping period was 54.4% in diabetic patients, whereas it was observed to be 60% in nondiabetic patients. Our diabetic patients exhibited lower saturation values compared to the determined “normal” values (despite higher blood pressure values being observed), which may indicate microvascular dysfunction [46]. This observation further strengthens the relevance of the use of neuromonitoring during carotid surgery, notedly in diabetic patients, as these pathological conditions may have exerted a negative impact on cognitive outcomes. Providing the most adequate level of anesthesia and keeping vital parameters in physiologic range may have highlighted the role in these patients of avoiding negative consequences.

We also aimed to explain the higher two-year mortality rate of diabetic patients. Elevated glucose levels are associated with oxidative stress and exert detrimental effects via numerous different cellular and biochemical pathways [47,48,49]. The greater degree of stenosis of the contralateral carotid artery observed in the diabetic group may offer less compensation for cerebral hemodynamics and may represent a sign of more severe atherosclerotic changes [50,51].

Diabetes mellitus was demonstrated to be significantly associated with the occurrence of long-term complications and mortality. In a multivariate model, only a carotid severity score by Blecha and the deterioration of cognitive function (as demonstrated by the MoCA test) were risk factors for this outcome. This result highlights and strengthens the importance of preserving cognitive function.

During the operation, diabetic patients required more vasopressors, and their blood pressure exhibited greater alterations and lower cerebral saturation values on both hemispheres, which may be influenced by cardiovascular autonomic neuropathy (CAN). Cardiovascular autonomic neuropathy is a microangiopathic complication of diabetes that affects the parasympathetic and sympathetic fibers that innervate the respiratory and cardiovascular systems [52,53]. The interaction of anesthesia and CAN may cause this hemodynamic instability. The interplay of them may explain the tendency toward lower cerebral saturation and a greater need for vasopressors.

Our results underscore the importance of neuromonitoring during carotid surgeries, especially in patients with diabetes mellitus, in favor of better postoperative outcomes.

As diabetic patients have a higher risk both for the occurrence of cognitive impairment and worse survival, accurate preoperative risk factor assessment and proper anesthesiologic care have a highlighted role in their management.

Limitations:

This was a single-center prospective study of patients who underwent surgery under general anesthesia. The small patient size limits our conclusions regarding MACCEs and mortality. Therefore, we used the score for adjustment in our cohort. Based on the preoperative and postoperative levels of HbA1c, the metabolic status of our diabetic patients was poorly controlled. As further limitation, we have to mention the lack of completeness analyzing those factors, which may have an impact on the alteration of cognition postoperatively.

## 5. Conclusions

In summary, diabetic patients are at increased risk for long-term cardiovascular and cerebrovascular complications and mortality, which are not demonstrated in the immediate perioperative period.

Cerebral tissue saturation values were significantly lower in the preclamping period among diabetic patients, and the lowest rSO_2_ values were also lower in this group. The observed hemodynamic instability and lower rSO_2_ values in diabetic patients may be linked to less cognitive improvement observed one year after the operation in the diabetic group.

## Figures and Tables

**Figure 1 biomedicines-13-02188-f001:**
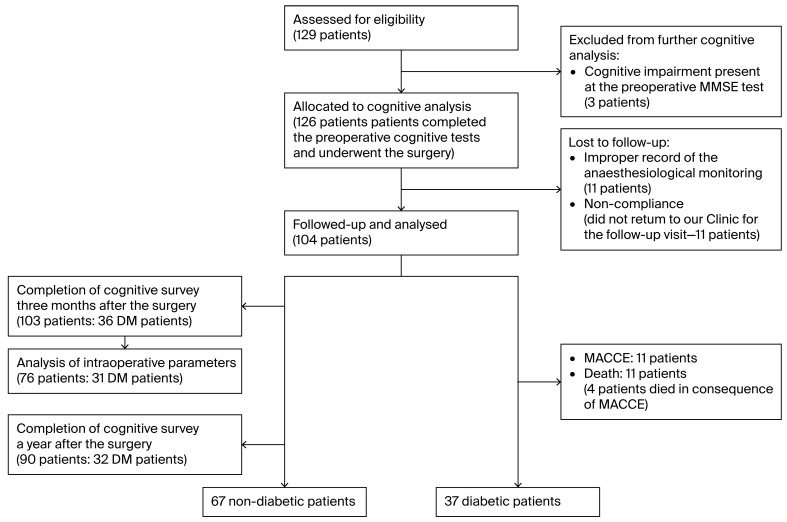
Consort flow diagram of the enrollment and follow-up.

**Figure 2 biomedicines-13-02188-f002:**
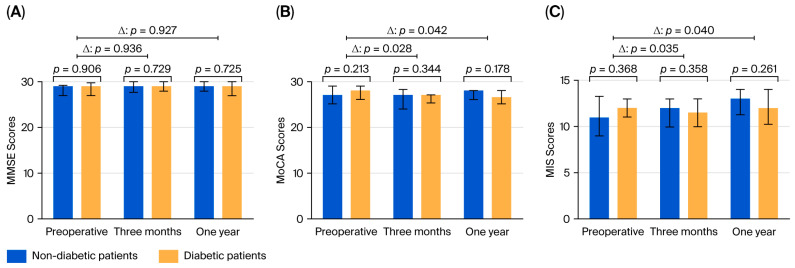
Results of the cognitive tests: (**A**) MMSE, (**B**) MoCA, and (**C**) MIS, showing the statistical signifi-cance of the difference between the groups and of the change from the preoperative level. MMSE: Mini-Mental State Examination; MoCA: Montreal Cognitive Assessment; MIS: Memory Index Score.

**Figure 3 biomedicines-13-02188-f003:**
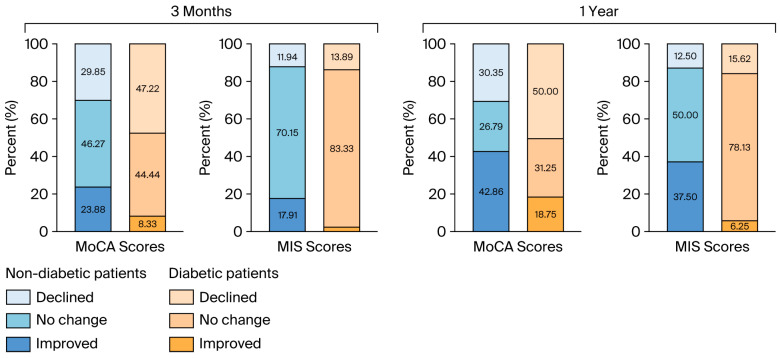
The proportions of the directions of cognitive changes based on the MoCA and MIS scores at 3 months and a year after surgery. MoCA: Montreal Cognitive Assessment; MIS: Memory Index Score.

**Figure 4 biomedicines-13-02188-f004:**
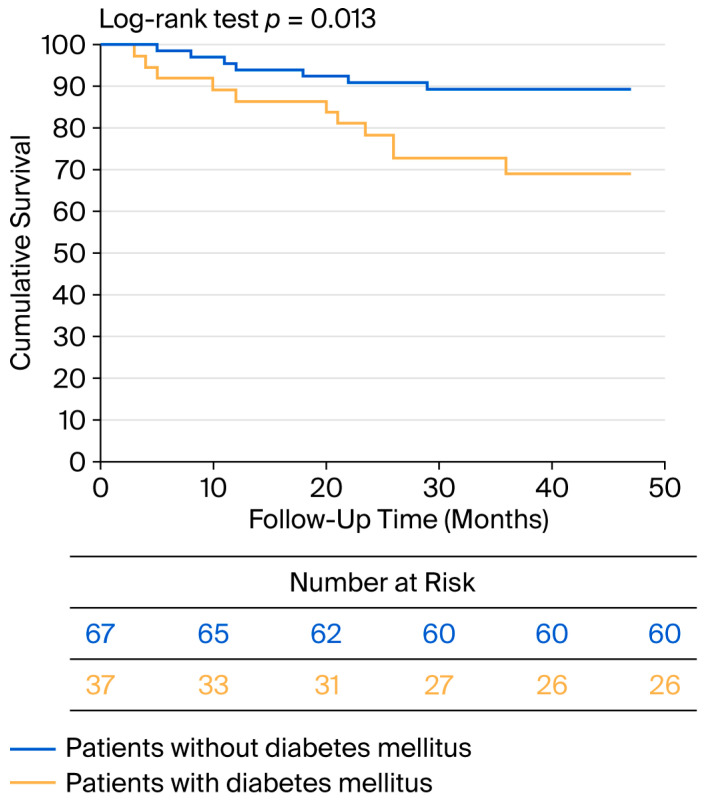
Kaplan–Meier curve for negative postoperative outcome in diabetic and nondiabetic patients.

**Figure 5 biomedicines-13-02188-f005:**
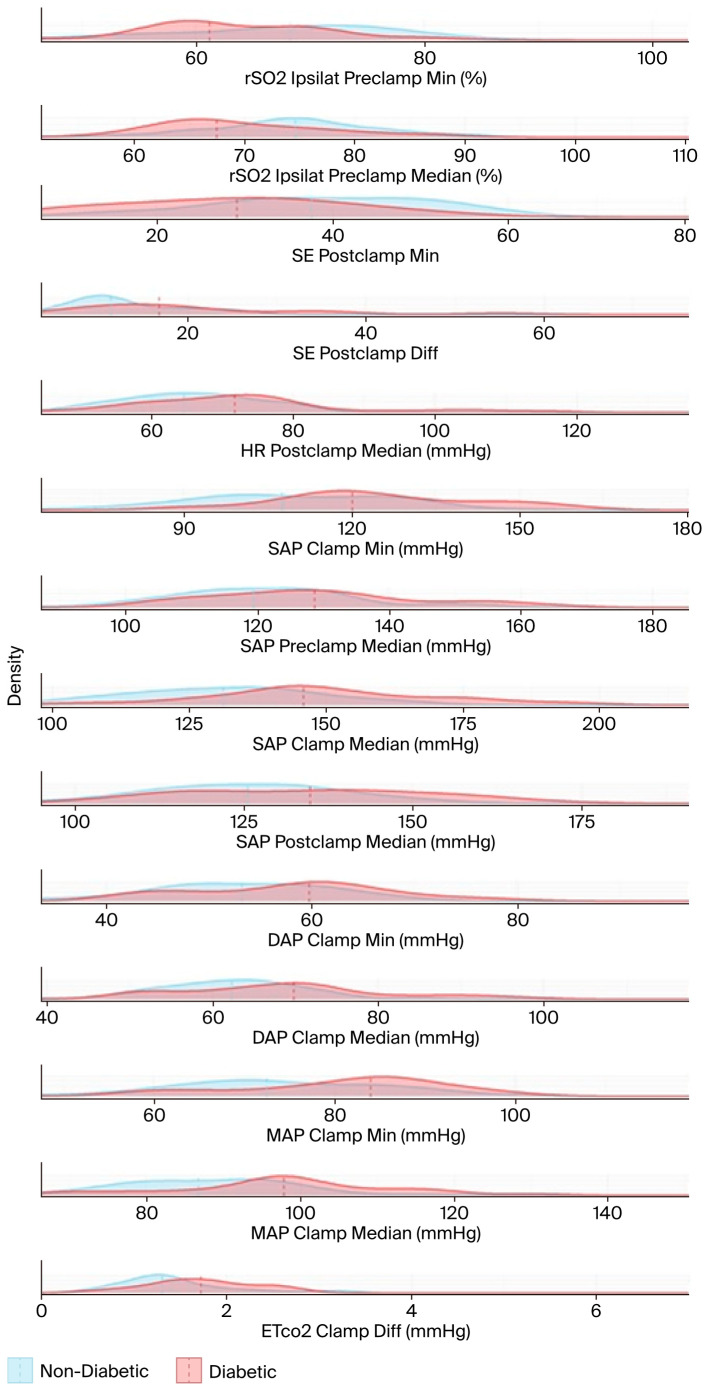
The density functions of the test variables used in the hypothesis tests, and the medians (dashed lines). The name of each test variable and its unit of measurement are displayed to the right of the corresponding density plot. DAP: diastolic arterial pressure; EtCO2: end-tidal carbon dioxide; HR: heart rate; MAP: mean arterial pressure; rSO2: regional cerebral oxygen saturation; SAP: systolic arterial pressure; SE: state entropy.

**Table 1 biomedicines-13-02188-t001:** Baseline characteristics of the two groups.

Characteristics	Patients Without Diabetes Mellitus (67 Patients)		Patients with Diabetes Mellitus (37 Patients)		*p* Value
Age (y)	70.81	7.42	69.89	6.73	0.539
Male	36	53.7	23	62.16	0.320
ASA	3	2–3	3	2–3	0.657
Vascular-POSSUM	18.82	3.73	19.81	3.86	0.210
BMI	28.40	24.69–29.38	27.96	25.65–30.94	0.318
Smoker	21	31.34	10	27.02	0.645
Alcohol	6	8.95	6	16.21	0.245
Anemia *	2	2.98	0	0	0.537
Hypertension	61	91.04	34	91.89	0.539
Ischemic Heart Disease	18	26.86	16	43.24	0.070
Congestive Heart Failure	4	5.97	10	27.02	0.003
Previous Stroke	16	23.9	7	18.91	0.606
Chronic Obstructive Pulmonary Disease	12	17.91	9	24.32	0.435
Preoperative Fasting Blood Glucose (mmol/L)	5.7	5.15–6.1	7.1	6.27–7.92	0.001
Hyperlipidemia	35	52.23	18	48.64	0.828
Cholesterol Level (mmol/L)	4.55	3.7–5.52	3.8	3.21–4.65	0.009
Triglyceride Level (mmol/L)	1.41	1.04–2.00	1.78	1.16–2.45	0.224
Statin Use	41	61.2	23	62.16	0.993
Chronic Renal Insufficiency **	1	1.5	5	13.5	0.012
Peripheral Arterial Disease (PAD)	14	20.89	12	32.43	0.193
Preoperative PAD Intervention	4	5.97	8	21.62	0.017
Previous CEA or Stent	2	2.98	4	10.81	0.101
Major Lower Extremity Amputation	0	0	2	5.40	0.124
Education (school years)	12.46	2.43	13.05	2.58	0.251
Low Financial State	2	2.98	2	5.40	0.539
Risk Score of Blecha et al.	2	1–4	5	3–7	0.001

Variables with normal distribution are presented as means and standard deviations, and nonnormally distributed data are presented as medians and interquartile ranges. Categorical data are presented as numbers and percentages. BMI: body mass index, CEA: carotid endarterectomy, PAD: peripheral arterial disease, Vascular-POSSUM: Vascular-Physiological and Operative Severity Score for enUmeration of Mortality and Morbidity. * Based on the definition of Blecha et al., preoperative hemoglobin <10 mg/dL. ** Based on the definition of Blecha et al., preoperative creatinine ≥1.3 mg/dL, no patient on dialysis.

**Table 2 biomedicines-13-02188-t002:** Operative parameters.

Characteristics	Patients Without Diabetes Mellitus (*n* = 67)		Patients with Diabetes Mellitus (*n* = 37)		*p* Value
Operated side (left)	30	44.8	20	54.05	0.297
Eversion endarterectomy (EEA)	54	80.6	18	48.65	0.001
Thromboendarterectomy (TEA)	13	19.40	19	51.35	0.001
Ipsilateral stenosis (%)	80	80–90	80	80–90	0.125
Contralateral stenosis (%)	40	0–50	55	22.5–78.75	0.012
Completeness of CoW	20	29.9	10	27.02	0.734
Shunt use	12	17.9	17	45.94	0.002
Shunt time (min)	30.5	20–41	34	28–39.5	0.347
Clamp time (min)	27.37	8.63	30.474	8.46	0.178

Nonnormally distributed data are presented as medians and interquartile ranges, and normally distributed data are presented as means and standard deviations. Categorical data are presented as numbers and percentages. CoW: circle of Willis.

**Table 3 biomedicines-13-02188-t003:** Multivariate Cox regression analysis for two-year mortality and major cardiovascular and cerebrovascular events.

Characteristics	AHR	95% C.I.		*p* Value
PNCD	4.477	1.289	15.550	0.018
Diabetes mellitus	1.351	0.489	3.738	0.562
Carotid severity score of Blecha et al.	1.435	1.165	1.767	0.001

PNCD: perioperative neurocognitive disorder, based on postoperative change in MoCA.

## Data Availability

The data that support the findings of this study are available from the corresponding author, [Andrea Székely], upon reasonable request.

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
