# Peer review of "Association of Diabetes with Greater Mid-Term Cognitive Decline After Carotid Surgery"

_biomedicines, 2025, doi:10.3390/biomedicines13092188_

Round 1

Reviewer 1 Report

Comments and Suggestions for Authors

The authors stated mild cognitive impairment of postoperative patients with DM.

Is there any difference of contents of cognitive impairment between each side of operation ?

Was vermal cognitive impairment more prominent in the patients with operations of the right side ?

Please the most plaucible pathogenesis of cognitive impairment after operation.

Reviewer 2 Report

Comments and Suggestions for Authors

This paper tackles an important topic—the relationship between diabetes, cerebral oxygenation, and mid-term cognitive outcomes following carotid endarterectomy (CEA). The combination of near-infrared spectroscopy (NIRS) with standardized cognitive testing is a particular strength. Below are a few suggestions aimed at improving clarity, tightening methodology, and increasing the overall impact of the work.

Introduction
It would help to briefly acknowledge earlier studies linking diabetes with postoperative cognitive decline in CEA patients. For example, “While diabetes is recognized as a risk factor for stroke after CEA [cite], its role in medium-term cognitive changes remains less clear.”
The narrative could also move more smoothly from diabetes to CEA to cognitive outcomes, perhaps by drawing a line between metabolic dysregulation and the risk of cerebral hypoperfusion during surgery.

Methods
Some procedural details could be made more explicit:

  • Patient selection: Clarify whether recruitment was consecutive or randomized.

  • Shunt use: State the criteria for insertion (e.g., “Shunts were placed in patients with contralateral stenosis >80% or an incomplete Circle of Willis.”).

  • Missing data: Explain how the 11 patients lost to follow-up were handled in the analysis.

  • Ethics: Double-check that the IRB approval number and date are correctly recorded (currently listed as “17/2019, 02/15/2019”).

Results

  • Figures: Make sure Figure 3 (showing proportions of cognitive change) appears in the final version.

  • Tables: Consider breaking up Table A1 into smaller, topic-specific tables for better readability, and add footnotes to Table 1 to clarify abbreviations such as PAD and CEA.

Discussion

  • Context: Position the results in relation to other work—e.g., “Our finding of lower rSO₂ in diabetic patients is consistent with [cite], which suggests that microvascular dysfunction could worsen intraoperative desaturation.”

  • Limitations: Note potential unmeasured confounders such as glycemic variability or sleep apnea.

  • Practical impact: Spell out how the findings might influence care, for example: “Routine NIRS monitoring in diabetic patients could help flag individuals at higher risk for cognitive decline, supporting closer follow-up.”

Language and style

  • Shortening complex sentences improves readability. For instance:
    Original – “The interaction of anesthesia and CAN may cause this hemodynamic instability and may explain the tendency toward lower cerebral saturation and a greater need for vasopressors.”
    Revised – “Anesthesia-related cardiovascular autonomic neuropathy (CAN) likely contributes to hemodynamic instability, reducing cerebral saturation and increasing vasopressor needs.”

  • Choose one consistent term—either “diabetic patients” or “DM patients”—and use it throughout.

References

  • Add recent literature (2023–2024) on diabetes and postoperative cognition.

  • Check that all DOIs and URLs (e.g., the IDF Diabetes Atlas) are active and accurate.

Overall
This is a well-executed study with a clear clinical question. With a few adjustments to strengthen rationale, clarify methods, and refine presentation, it could make a meaningful contribution to the literature.

Comments on the Quality of English Language

See comment above.
